# A Law of Iterated Logarithm for Multi-Agent Reinforcement Learning

**Gugan Thoppe**
Computer Science and Automation
Indian Institute of Science
Bengaluru, Karnataka 560012, India
gthoppe@iisc.ac.in

**Bhumesh Kumar**
Electrical and Computer Engineering
University of Wisconsin at Madison
Madison, WI 53706, USA
bkumar@wisc.edu

## Abstract

In Multi-Agent Reinforcement Learning (MARL), multiple agents interact with a common environment, as also with each other, for solving a shared problem in sequential decision-making. It has wide-ranging applications in gaming, robotics, finance, etc. In this work, we derive a novel law of iterated logarithm for a family of distributed nonlinear stochastic approximation schemes that is useful in MARL. In particular, our result describes the convergence rate on almost every sample path where the algorithm converges. This result is the first of its kind in the distributed setup and provides deeper insights than the existing ones, which only discuss convergence rates in the expected or the CLT sense. Importantly, our result holds under significantly weaker assumptions: neither the gossip matrix needs to be doubly stochastic nor the stepsizes square summable. As an application, we show that, for the stepsize $n^{-\gamma}$ with $\gamma \in (0, 1)$, the distributed TD(0) algorithm with linear function approximation has a convergence rate of $\mathcal{O}(\sqrt{n^{-\gamma} \ln n})$ a.s.; for the $1/n$ type stepsize, the same is $\mathcal{O}(\sqrt{n^{-1} \ln \ln n})$ a.s. These decay rates do not depend on the graph depicting the interactions among the different agents.

## 1 Introduction

Can a machine train itself in the same way an infant learns to sit up, crawl, and walk? That is, can a device interact with the environment and figure out the action sequence required to complete a given task? The study of *algorithms* that enable such decision-making is what the field of Reinforcement Learning (RL) is all about [40]. In contrast, the *mathematics* needed to analyze such schemes is what forms the focus in Stochastic Approximation (SA) theory [2, 4]. More generally, SA refers to an iterative scheme that helps find zeroes or optimal points of a function, for which only noisy evaluations are possible. In this work, we analyze a family of Distributed Stochastic Approximation (DSA) algorithms [24] that is useful in Multi-Agent Reinforcement Learning (MARL) [22, 51].

In the MARL framework, we have multiple agents or learners that continually engage with a shared environment: the agents pick local actions, and the environment responds by transitioning to a new state and giving each agent a different local reward. Additionally, the agents also *gossip* about local computations with each other. The goal of the agents is to cooperatively find action policies that maximize the collective rewards obtained over time. Algorithms useful in this endeavor have found empirical success in domains as diverse as gaming [28], robotics [29], autonomous driving [35], communication networks [21], power grids [34], and economics [19]. However, theoretical analyses of such schemes are still very minimal, and this is what this paper aims to address.

For the purpose of analysis, MARL methods are often viewed as special cases of DSA algorithms. The archetypical form of a DSA scheme with $m$ distributed nodes can be described as follows. Let $\mathcal{G}$

be a directed graph representing the connections between these nodes, and $W \equiv (W_{ij}) \in [0, 1]^{m \times m}$ a matrix whose $ij$-th entry denotes the strength of the edge $j \to i$. It is assumed that $W$ is compatible with $\mathcal{G}$, i.e., $W_{ij} > 0$ only if $j \to i \in \mathcal{G}$. Then, at agent $i$, the above scheme (written as row vectors) has the update rule

$$x_{n+1}(i) = \sum_{j \in \mathcal{N}_i} W_{ij} x_n(j) + \alpha_n[h_i(x_n) + M_{n+1}(i)], \qquad n \geq 0, \tag{1}$$

where $x_n \in \mathbb{R}^{m \times d}$ is the joint estimate of the solution at time $n$, its $j$-th row, i.e., $x_n(j)$ denotes[1] the estimate obtained at agent $j$, $\mathcal{N}_i$ represents the set of in-neighbors of node $i$ in $\mathcal{G}$, $\alpha_n$ is the stepsize, $h_i : \mathbb{R}^{m \times n} \to \mathbb{R}^d$ is the driving function at agent $i$, and $M_{n+1}(i) \in \mathbb{R}^d$ is the noise in its evaluation at time $n$. This update rule has two parts: a weighted average of the estimates obtained by gossip and a refinement based on local computations. Clearly, the joint update rule of all the agents is

$$x_{n+1} = W x_n + \alpha_n[h(x_n) + M_{n+1}], \tag{2}$$

where $M_{n+1}$ is the $m \times d$ matrix whose $i$-th row is $M_{n+1}(i)$, and $h$ is the function that maps $x \in \mathbb{R}^{m \times d}$ to the $m \times d$ matrix whose $i$-th row is $h_i(x)$.

Two important points about the above framework are as follows: i.) we allow $h_i$ to be a function of all of $x_n$ and not just of $x_n(i)$, as is commonly assumed, and ii.) the computations at different nodes in the above setup run synchronously on a common clock.

**Related Work**: We now give a summary of relevant theoretical results from the DSA and MARL literature. For ease of discussion, we categorize them into i.) asymptotic and ii.) finite-time results.

The asymptotic ones mainly concern almost sure (a.s.) convergence [42, 3, 26, 24, 15, 52, 50, 39, 18]. The first four papers here provide convergence guarantees for a broad family of nonlinear DSA algorithms. The other articles also do the same, but in context of specific MARL schemes such as distributed Q-learning, distributed actor-critic methods, distributed TD methods, and their off-policy variations. Two other kinds of asymptotic results also exist in the literature. The first is the CLT shown in [27] for the average of estimates obtained at different nodes in a generic DSA scheme. The other is the convergence in mean result obtained in [48] for a distributed policy gradient method.

Finite-time literature, in contrast, majorly talks about expectation bounds. Assuming there exists a unique $x_*$ that solves $\sum_{i=1}^m h_i(x) = 0$, these results describe the rate at which $\mathbb{E}\|x_n - x_*\|$ decays with $n$. Notable contributions for DSA here are [49, 44]. Compared to ours, these look at a slightly different setup: the measurement noise at each node has a Markov component in place of a martingale difference term. In this new setup, [49] shows that, for any sufficiently small but constant stepsize $\epsilon$, the expected error decreases linearly to a ball of radius $\mathcal{O}(\sqrt{\epsilon \ln(1/\epsilon)})$. On the other hand, [44] deals with the case where $h$ is additionally non-convex and shows that $\mathbb{E}\|x_n - x_*\| = \mathcal{O}(n^{-1/4}\sqrt{\ln n})$, which is comparable to the best known bound in the centralized setting.

Expectation bounds in the MARL framework primarily concern policy evaluation methods [10, 11, 37, 5]. The first three papers here deal with the distributed TD(0) method. These show that a result similar to the one in [49] holds for this method under constant stepsizes. In contrast, when $\alpha_n$ is of the $1/n$ type, it is proven that $\mathbb{E}\|x_n - x_*\| = \mathcal{O}(1/\sqrt{n})$. Similar bounds have also been derived in [5] for two distributed variants of the TDC method. There are also some other works that derive finite-time bounds [45, 9, 46, 54, 13, 36, 33, 53], but we do not discuss them in this paper since the algorithms proposed there do not fit the update rule given in (2).

The different finite-time results, as also the asymptotic CLT, do provide insights into the rate at which an iterative method converges. However, there are some significant issues with these studies. First, except [27], all others require the gossip matrix to be doubly stochastic, at least in the mean. While this assumption simplifies the analysis, it also severely restricts the communication protocol choices. In fact, as pointed out in [27], this condition even limits the use of a natural broadcast node, one that transmits its local estimate to all the neighbors without expecting all of them to respond. Second, these works only talk about convergence rates in the expected or the CLT sense. By their very nature, these results do not reveal much about the decay rates along different sample paths. Finally, all current results, including the ones on convergence, only apply to constant or square-summable stepsizes. Nothing is known about the slowly-decaying non-square-summable

---

[1] By default, all our vectors are row vectors. We use $'$ for conjugate transpose.

ones, which are generally preferable since they give similar benefits as constant stepsizes and, often, also guarantee convergence. Note that such issues also plague much of the distributed stochastic optimization literature [47, 38, 20, 16, 32, 31].

**Key Contributions**: The highlights of this work are as follows.

1. **Law of Iterated Logarithm (LIL)**: We derive a novel law of iterated logarithm for the DSA scheme given in (2). That is, for a suitably defined $x_*$, we show that $\limsup[\alpha_n \ln t_{n+1}]^{-1/2}\|x_n - x_*\| \leq C$ a.s. on every sample path in the event

$$\mathcal{E}(x_*) := \{x_n \to x_*\}. \tag{3}$$

Here, $C \geq 0$ is some constant[2] and $t_n = \sum_{k=0}^{n-1} \alpha_k$. Also, the norm that we work is the operator norm. In particular, for any $x \in \mathbb{C}^{m \times d}$,

$$\|x\| := \sup_{u \in \mathbb{C}^m, v \in \mathbb{C}^d} \{|uxv'| : \|u\| = \|v\| = 1\}. \tag{4}$$

This result is the first of its kind in the distributed setup. Further, as discussed in Remark 2.3 later, it provides deeper insights about the asymptotic behavior of $(x_n)_{n \geq 0}$ than other existing results, which only discuss convergence rates in the expected or the CLT sense.

2. **Analysis and Gossip Matrix**: The above result is obtained via a new approach we develop here for analyzing DSA schemes. Let $\pi \in \mathbb{R}^m$ be such that $\pi W = \pi$ and let

$$Q := \mathbb{I} - \mathbf{1}'\pi, \tag{5}$$

where $\mathbf{1} \in \mathbb{R}^m$ denotes the vector of all ones. Then an outline of our approach is that we express $x_n - x_*$ as a sum of $\mathbf{1}'\pi(x_n - x_*)$ and $Qx_n$ and, thereafter, analyze each summand by treating its update rule as a separate SA scheme. This contrasts the usual approach (e.g., [27, 10, 11]) where the error is split into $(\mathbf{1}'\mathbf{1}/m)(x_n - x_*)$ and $(\mathbb{I} - \mathbf{1}'\mathbf{1}/m)x_n$. In fact, this is the main reason why, unlike other existing results, ours does not require that the gossip matrix be doubly stochastic.

3. **Concentration Inequality and Stepsizes**: We also improve upon an existing concentration result ([12, Corollary 6.4.25]) for a sum of martingale differences; see Lemma 4.6. Specifically, by modifying the original proof from [12], we show that the result stated there actually holds under a broader set of conditions. The key benefit of this is that, unlike other related results, our LIL result does not require that the stepsize sequence be square-summable.

4. **MARL Application**: We use our theory to prove a law of iterated logarithm for the distributed TD(0) algorithm with linear function approximation. This is the first such result in MARL.

**Contents**: The rest of the paper is structured as follows. In Section 2, we formally state our main result along with all the assumptions needed. We also pinpoint the new insights that our result provides. In Section 3, we give a demonstration of how our result can be applied in the MARL setup. In particular, there we talk about the distributed TD(0) algorithm with linear function approximation and prove that it indeed satisfies all the assumptions of our main result. Section 4 has two parts. In the first part, we state some key intermediate lemmas and then use the same to derive our main result. The latter part, in contrast, focuses on proofs of these intermediate results; note that we only sketch their proofs here and leave the details to the appendix. Finally, in Section 5, we conclude with a summary of our findings and discuss some interesting future directions.

## 2 Assumptions and Main Result

Throughout this work, we assume that the following four technical assumptions, i.e., $\mathcal{A}_1, \ldots, \mathcal{A}_4$, hold for the DSA scheme in (2).

$\mathcal{A}_1$. **Property of the Gossip Matrix**: *W is an irreducible aperiodic row stochastic matrix*.

This condition implies there exists a unique vector $\pi \in \mathbb{R}^m$ such that

$$\pi W = \pi. \tag{6}$$

---

[2]Throughout, $C$ denotes a generic constant. Its value could be different each time it is used; in fact, it could be different even in the same line.

Accordingly, based on [24, Theorem 1], one would expect (2) to eventually converge to an invariant set of the $m$-fold product of the $d$-dimensional ODE

$$\dot{y}(t) = \sum_{i=1}^{m} \pi_i h_i(\mathbf{1}'y(t)) = \pi h(\mathbf{1}'y(t)). \tag{7}$$

By an $m$-fold product, we refer to the dynamics in $\mathbb{R}^{m \times d}$ where each row individually satisfies (7). A natural invariant set of this dynamics is $\mathcal{S} := \{\mathbf{1}'y : y \in \mathbb{R}^d\} \subset \mathbb{R}^{m \times d}$. With this in mind, let

$$x_* = \mathbf{1}'y_* \in \mathcal{S}, \tag{8}$$

where $y_* \in \mathbb{R}^d$ is an asymptotically stable equilibrium of (7). Notice that we don't assume $y_*$ to be the only attractor of this ODE.

We remark that our main result concerns the behavior of the DSA scheme on the event $\mathcal{E}(x_*)$, where $x_*$ is as defined above and $\mathcal{E}(x_*)$ is as defined in (3).

$\mathcal{A}_2$. **Nature of $h$ near $x_*$**: *There exists a neighbourhood $\mathcal{U}$ of $x_*$ such that, for $x \in \mathcal{U}$,*

$$h(x) = -\mathbf{1}'\pi(x - x_*)A + \mathbf{1}'\pi f_1(x) + Q(B + f_2(x)), \tag{9}$$

*where $A \in \mathbb{R}^{d \times d}$ is such that $yAy' > 0$ for all $y \neq 0$, $B \in \mathbb{R}^{m \times d}$ is some constant matrix, $f_2 : \mathcal{U} \to \mathbb{R}^{m \times d}$ is some arbitrary continuous function, while $f_1 : \mathcal{U} \to \mathbb{R}^{m \times d}$ is another continuous function that additionally satisfies*

$$\|\mathbf{1}'\pi f_1(x)\| = \mathcal{O}(\|\mathbf{1}'\pi(x - x_*)\|^a), \qquad \text{as } x \to x_*, \tag{10}$$

*for some $a > 1$.*

Note that $\mathcal{A}_2$ is a generalization of **Assumption (A1)** in [30]. As in [30], this also is local in nature: it only prescribes a specific behavior for $h$ close to $x_*$. Furthermore, this condition ensures that the driving function $\pi h(x)$ in (7) equals $-\pi(x - x_*)A + \pi f_1(x)$; the first term is the linear part while the second term represents the nonlinear portions. Separately, observe that $Qh(x) = Q(B + f_2(x))$. This plays no role in (7); hence, conditions on $B$ and $f_2$ are minimal. We now construct a family of examples to show that $\mathcal{A}_2$ broadly holds. The simplest member in this family is $h(x) = B - xA$, where $B$ and $A$ are as defined above[3]. Clearly, if $b(i)$ and $x(i)$ are the $i$-th rows of $B$ and $x$, respectively, then the $i$-component function here is $h_i(x) = b(i) - x(i)A$. The fact that the scaling matrix $A$ is the same for each $i$ is crucial for $\mathcal{A}_2$ to hold. Also, observe that this function does not depend on $\pi$. The other members of the family are obtained by adding various $\pi$-dependent nonlinear perturbations to this simple setup, i.e., by making different choices[4] for $f_1$ and $f_2$.

$\mathcal{A}_3$. **Stepsize Behavior**: *There exists some decreasing positive function $\alpha$ defined on $[0, \infty)$ such that the stepsize $\alpha_n = \alpha(n)$. Further, $\alpha$ is either of Type 1 or Type $\gamma$.*

(a) *Type 1: $\alpha(n) = \alpha_0/n$ for some $\alpha_0 > 1/(2\lambda_{\min})$, where*

$$\lambda_{\min} := \min\{\mathscr{R}(\lambda) : \lambda \in \text{spectrum}(A)\} \tag{11}$$

*with $\mathscr{R}(\lambda)$ denoting the real part of $\lambda$;*

(b) *Type $\gamma$: The function $\alpha$ is differentiable and its derivative varies regularly with exponent $-1 - \gamma$, where $0 < \gamma < 1$.*

The regularly varying condition above implies that $\left|\frac{d\alpha(x)}{dx}\right| = x^{-\gamma-1}L(x)$ for some slowly varying function $L$, e.g., $L(x) = C$ for some $C > 0$, or $L(x) = (\ln x)^\eta$ for some $\eta \in \mathbb{R}$. Thus, examples of $\alpha_n$ here include $Cn^{-\gamma}$ and $n^{-\gamma}(\ln n)^\eta$, which are non-square-summable for $\gamma \in (0, 1/2]$.

$\mathcal{A}_4$. **Noise Attributes**: *With $\mathcal{F}_n = \sigma(x_0, M_1, \ldots, M_n)$, and $\mathcal{E}(x_*)$, as in (3), the following hold.*

(a) $\mathbb{E}(M_{n+1}|\mathcal{F}_n) = 0$ *a.s.*

---

[3] A verification of all the conditions mentioned in $\mathcal{A}_2$ for $h(x) = B - xA$ has been done in Section 3; there we also discuss usefulness of this function in the context of policy evaluation in MARL.

[4] For example, we can let $f_1(x) = g_1(\pi(x - x_*))$ and $f_2(x) = -xA + g_2(x)$, where $g_1 : \mathbb{R}^d \to \mathbb{R}^{m \times d}$ is such that, as $z \to 0$, $\|g_1(z)\| = \mathcal{O}(\|z\|^a)$ for some $a > 1$, and $g_2$ is an arbitrary continuous function.

(b) *There exists $C \geq 0$ such that $\|QM_{n+1}\| \leq C\left(1 + \|Q(x_n - x_*)\|\right)$ a.s. on $\mathcal{E}(x_*)$.*

(c) *There is a non-random symmetric positive semi-definite matrix $M \in \mathbb{R}^{d \times d}$ such that*

$$\lim_{n \to \infty} \mathbb{E}(M'_{n+1}\pi'\pi M_{n+1} \mid \mathcal{F}_n) = M \quad \text{a.s. on } \mathcal{E}(x_*). \tag{12}$$

(d) *There exists $b > 2$ such that $\sup_{n \geq 0} \mathbb{E}(\|\pi M_{n+1}\|^b | \mathcal{F}_n) < \infty$ a.s. on $\mathcal{E}(x_*)$.*

These noise conditions are extensions of the standard assumptions in the SA literature [30, 25, 4].

Our main result can now be stated as follows. This generalizes Theorem 1 from [30].

**Theorem 2.1** (**Main Result: Law of Iterated Logarithm**)**.** *Suppose $\mathcal{A}_1, \ldots, \mathcal{A}_4$ hold and $\gamma > 2/b$ if $\alpha$ is of Type $\gamma$. Then, there exists some deterministic constant $C \geq 0$ such that*

$$\limsup [\alpha_n \ln t_{n+1}]^{-1/2} \|x_n - x_*\| \leq C \quad \text{a.s. on } \mathcal{E}(x_*).$$

This result is called a law of iterated logarithm since its proof crucially relies on Lemma 4.6, which indeed is a law of iterated logarithm for a sum of scaled martingale differences. We end this section with some important comments about our main result.

**Remark 2.2.** *Our result shows that, a.s. on $\mathcal{E}(x_*)$, $\|x_n - x_*\|$ is $\mathcal{O}(\sqrt{n^{-1} \ln \ln n})$ in the Type 1 case, and $\mathcal{O}(\sqrt{n^{-\gamma} \ln n})$ in the Type $\gamma$ case. Note that, since we require $\gamma > 2/b$, our result applies for smaller values of $\gamma$, only if $\mathcal{A}_4$.(d) holds for a sufficiently large $b$.*

**Remark 2.3.** *Our result provides deeper insights than the convergence rates that exist in the DSA/MARL literature. For this discussion, we suppose $\mathbb{P}\{\mathcal{E}(x_*)\} = 1$. As mentioned in Section 1, the existing results are of two kinds: finite-time expectation bounds and the CLT. Indeed a finite-time bound has several benefits and is not directly comparable to an asymptotic result. Nevertheless, an expectation bound only describes the average behavior, while ours characterizes the decay rate on almost every sample path. In fact, if we compare just the decay rate obtained in our result in the Type 1 case with that obtained in [10, 11], which show $\mathbb{E}\|x_n - x_*\| = \mathcal{O}(\sqrt{\ln n}/\sqrt{n})$, then ours is tighter (it has $\ln \ln n$ in place of $\ln n$). Furthermore, while a CLT can at the best show that $\limsup \alpha_n^{-1/2} \|x_n - x_*\| = \infty$ a.s., our result is more precise in stating that the expression becomes bounded if it is divided by an additional $\sqrt{\ln t_{n+1}}$ term.*

## 3 Application to Reinforcement Learning

We apply our result here to a variant of the distributed TD(0) algorithm [10, 11] with linear function approximation. This method is useful for policy evaluation in MARL. The discussion here is divided into the following three parts: i.) setup, ii.) objective and algorithm, and iii.) analysis.

**Setup**: We consider a distributed system of $m$ agents modeled by a Markov Decision Process. This can be characterized by the tuple $(\mathcal{S}, \{\mathcal{U}_i\}, \mathcal{P}, \{\mathcal{R}_i\}, \gamma, \mathcal{G})$. Here, $\mathcal{S} = \{1, \ldots, L\}$ is the global state space, $\mathcal{U}_i$ and $\mathcal{R}_i$ are the set of actions and the reward function at agent $i$, respectively, $\mathcal{P}$ describes the transition probabilities, $\gamma$ is the discount factor, and $\mathcal{G} \equiv (\mathcal{V}, \mathcal{E})$ is a directed graph that represents the connectivity structure among the $m$ agents.

Let $\mathcal{N}_i$ and $W$ be as in (1). We assume that this $W$ satisfies the conditions in $\mathcal{A}_1$. Then, for this matrix, there is a unique vector $\pi \equiv (\pi_i)$ satisfying (6).

Let $\mu_i$ be the stationary policy of agent $i$ and let $\mu \equiv (\mu_i)$. Also, let $\mu(a|s) = \prod_i \mu_i(a_i|s)$ be the probability for choosing the joint action $a \equiv (a_i) \in \prod_i \mathcal{U}_i$. This policy $\mu$ then induces a Markov chain on $\mathcal{S}$, which we assume is aperiodic and irreducible. Therefore, it also has a unique stationary distribution and we denote the same by $\varphi \in \mathbb{R}^L$.

At each step, the above system evolves as follows. First, each agent $i$ sees the current state $s$ and applies an action $a_i \in \mathcal{U}_i$ sampled from $\mu_i(\cdot|s)$. Based on the joint action $a$, the system then moves to a new state $\tilde{s}$. Equivalently, the joint action $a$ and the state $\tilde{s}$ can be seen as samples of $\mu(\cdot|s)$ and $\mathcal{P}(\cdot|s, a)$, respectively. Finally, each agent $i$ receives an instantaneous reward $\mathcal{R}_i(s, a, \tilde{s})$.

**Objective and Algorithm**: The goal of the multi-agent system is to cooperatively estimate the value function $J^\mu \in \mathbb{R}^L$ corresponding to $\mu$. This is defined as the solution to the Bellman equation

$$J^\mu(s) = \mathbb{E}\left[ \sum_i \pi_i \mathcal{R}_i(s, a, \tilde{s}) + \gamma J^\mu(\tilde{s}) \right], \quad s \in \mathcal{S},$$

where the expectation is over $a \sim \mu(\cdot|s)$ and $\tilde{s} \sim \mathcal{P}(\cdot|s,a)$. This expression differs from the ones in [10, 11], in that, the coefficients $\pi_i$ here is replaced by $1/m$ there. When $L$ is large, estimating $J^\mu$ directly is intractable. An alternative then is to make use of linear function approximation. That is, for some $d$, choose a feature matrix $\Phi \in \mathbb{R}^{L \times d}$ with full column rank. And then, with $\phi(s)$ denoting the $s$-th row of $\Phi$, try and find a $\theta \in \mathbb{R}^d$ such that $J^\mu(s) \approx \phi(s)\theta'$ for all $s \in \mathcal{S}$.

The distributed TD(0) algorithm is helpful in this latter context. Let $(s_n, a_n, \tilde{s}_n)$, $n \geq 0$, be IID[5] samples of $(s, a, \tilde{s})$, where $s \sim \varphi(\cdot), a \sim \mu(\cdot|s)$, and $\tilde{s} \sim \mathcal{P}(\cdot|s,a)$. Then, at agent $i$, this distributed algorithm has the update rule:

$$\theta_{n+1}(i) = \sum_{j \in \mathcal{N}_i} W_{ij}\theta_n(j) + \alpha_n(b_n(i) - \theta_n(i)A_n), \tag{13}$$

where $b_n(i) = \mathcal{R}_i(s_n, a_n, \tilde{s}_n)\phi(s_n) \in \mathbb{R}^d$ and $A_n = \phi'(s_n)\phi(s_n) - \gamma\phi'(\tilde{s}_n)\phi(s_n) \in \mathbb{R}^{d \times d}$.

**Analysis**: We first express the update rule given in (13) for different $i$ in the standard DSA form. Let[6] $A = \mathbb{E}[A_n]$ and $B = \mathbb{E}[B_n]$, where $B_n \in \mathbb{R}^{m \times d}$ is the matrix whose $i$-th row is $b_n(i)$. Both $A$ and $B$ do not depend on $n$ since $(s_n, a_n, \tilde{s}_n)$, $n \geq 0$, is IID. Next, for $n \geq 0$, let $x_n \in \mathbb{R}^{m \times d}$ be the matrix whose $i$-th row is $\theta_n(i)$. Then, (13) for different $i$ can be jointly written as shown in (2) for

$$h(x) = B - xA \qquad \text{and} \qquad M_{n+1} = (B_n - B) - x_n(A_n - A). \tag{14}$$

Next, we look at the limiting ODE given in (7). In our case, this has the form $\dot{y}(t) = \pi B - y(t)A$. Now, $A$ is known to be positive definite [40], i.e., $\theta A\theta' > 0$ for all $\theta \neq 0$. Hence, it is invertible and the real parts of all its eigenvalues are positive, i.e., $-A$ is Hurwitz stable. This then shows that $\theta_* = \pi BA^{-1}$ is the unique globally asymptotically stable equilibrium for the above ODE.

We now verify the assumptions stated in Section 2. $\mathcal{A}_1$ trivially holds due to assumptions on $W$. For $\mathcal{A}_2$, let $f_1(x) = 0$ and $f_2(x) = -xA$ for $x \in \mathbb{R}^{m \times d}$. Further, let $x_* = \mathbf{1}'\theta_*$. Then, $h(x) = -\mathbf{1}\pi(x - x_*)A + (\mathbb{I} - \mathbf{1}\pi)(B + f_2(x))$, as desired. In order to satisfy $\mathcal{A}_3$, we simply choose a stepsize sequence that fulfills one of the criteria mentioned there.

It now only remains to establish $\mathcal{A}_4$. Let $\mathcal{F}_n$ be as defined there. Then, part (a) follows from the definitions of $A$ and $B$ and the fact that $(s_n, a_n, \tilde{s}_n)$ is independent of the past. On the other hand, part (b) can be shown by building upon the arguments used in the proof of [7, Lemma 5.1]. Next observe that, since $(s_n, a_n, \tilde{s}_n)$ is independent of the past, the only quantity that is random in $\mathbb{E}[M'_{n+1}\pi'\pi M_{n+1}|\mathcal{F}_n]$ is $x_n$. Also, trivially, $\mathbb{E}[M'_{n+1}\pi'\pi M_{n+1}|\mathcal{F}_n]$ is a symmetric positive semi-definite matrix. Therefore, on the event $\mathcal{E}(x_*)$, it is easy to see that part (c) holds as well. Finally, notice that $\|\pi M_{n+1}\| \leq C(1 + \|x_n - x_*\|)$ for some $C \geq 0$; this follows as in part (b) above. Hence, on $\mathcal{E}(x_*)$, $\sup_{n \geq 0} \mathbb{E}[\|\pi M_{n+1}\|^b|\mathcal{F}_n] < \infty$ a.s. for any $b \geq 0$. This verifies part (d).

Thus, Theorem 2.1 holds for the distributed TD(0) algorithm with linear function approximation.

## 4 Theoretical Analysis: Proof of the Main Result

We now turn to the technical details of our analysis. With $Q$ as in (5) and $x_*$ as in (8), observe that $\mathbf{1}'\pi x_* = x_*$ and, hence, $x_n - x_* = \mathbf{1}'\pi(x_n - x_*) + Qx_n$. We refer to the first term in this decomposition as the agreement component of the error and the second as the disagreement component. This decomposition differs from the standard approaches [10, 11, 27], wherein $x_n - x_*$ is split into $(\mathbf{1}'\mathbf{1}/m)(x_n - x_*)$ and $(\mathbb{I} - (\mathbf{1}'\mathbf{1}/m))x_n$. In fact, the success of our approach strongly hinges on this novel error decomposition.

The rest of the section is organized as follows. We first state our bounds for the two terms in our decomposition. Using these bounds, we then provide a formal proof for Theorem 2.1. Thereafter, we sketch the proofs of these intermediate bounds, leaving the details to the appendix.

---

[5]The IID assumption is standard in literature [7, 23, 41, 43] and is needed to ensure that the update rule only has martingale noise. Otherwise, the update rule will additionally have Markovian noise, the analysis of which is beyond the scope of this paper. The good news though is that, as shown in previous works [14, 10, 11], the asymptotic behaviours with and without the Markovian noise are often similar.

[6]The matrix $A$ is usually defined to be the transpose of the expression we use.

**Lemma 4.1.** *(Agreement Error) Almost surely on $\mathcal{E}(x_*)$,*

$$\limsup_{n\to\infty} \frac{\|\mathbf{1}'\pi(x_n - x_*)\|}{\sqrt{\alpha_n \ln t_{n+1}}} \leq C, \tag{15}$$

*where $C \geq 0$ is some deterministic constant.*

**Lemma 4.2.** *(Disagreement Error) Let $\delta > 0$. Then,*

$$\|Qx_n\| = \mathcal{O}\left(\alpha_n(\ln n)^{1+\delta}\right) \quad \textit{a.s. on } \mathcal{E}(x_*). \tag{16}$$

**Remark 4.3.** *Up to logarithmic factors, the rate at which the disagreement error decreases is the square of the rate at which the agreement error decreases. Thus, the overall convergence rate is essentially dictated by the agreement component of the error.*

With these two ingredients at hand, our main result is arrived at via the following short calculation.

*Proof of Theorem 2.1.* Observe that

$$\|x_n - x_*\| \leq \|\mathbf{1}'\pi(x_n - x_*)\| + \|Qx_n\|.$$

Also, $\ln t_{n+1}$ is $O(\ln n)$ and $O(\ln\ln n)$ in the Type $\gamma$ and Type 1 cases, respectively. The desired result is now easy to see from Lemmas 4.1 and 4.2. $\qquad\square$

### 4.1 Bound on agreement error $\|\mathbf{1}'\pi(x_n - x_*)\|$

We first focus on the details of our analysis for the first ingredient, i.e., the agreement error. Let

$$\psi_{n+1} := \sum_{k=0}^{n} \alpha_k \mathbf{1}'\pi M_{k+1} e^{-(t_{n+1}-t_{k+1})A}, \quad n \geq 0, \tag{17}$$

and

$$\Delta_n := \mathbf{1}'\pi(x_n - x_*) - \psi_n, \quad n \geq 0. \tag{18}$$

Clearly, to prove Lemma 4.1, it suffices to obtain bounds on the rate at which $\|\psi_n\|$ and $\|\Delta_n\|$ decay. These bounds are stated below. Note that these results are generalizations of Lemmas 1 and 3 from [30]. Specifically, the results there focused on one-timescale stochastic approximation, ours on the other hand handles the distributed case. Furthermore, the quantities of interest here, e.g, $\psi_n, \Delta_n$, are matrix-valued, unlike the ones in [30] which were vector-valued.

**Lemma 4.4.** *Let $b$ be as in $\mathcal{A}_4$. Suppose that either $\alpha$ is of Type 1 or that $\alpha$ is of Type $\gamma$ with $\gamma > 2/b$. Then, there exists some deterministic constant $C \geq 0$ such that*

$$\limsup_{n\to\infty} (\alpha_n \ln t_{n+1})^{-1/2} \|\psi_{n+1}\| \leq C \quad \textit{a.s. on } \mathcal{E}(x_*) \tag{19}$$

**Lemma 4.5.** *Suppose $\mathcal{A}_2$, $\mathcal{A}_3$ and $\mathcal{A}_4$ hold. Then, for any $\lambda \in (0, \lambda_{\min})$,*

$$\|\Delta_n\| = \mathcal{O}\left(\max\left\{ e^{-\lambda\sum_{k=0}^{n}\alpha_k}, \sum_{j=0}^{n}\alpha_j e^{-\lambda\sum_{k=j+1}^{n}\alpha_k}[\alpha_j\|\psi_j\| + \|\psi_j\|^a]\right\}\right) \tag{20}$$

*a.s. on $\mathcal{E}(x_*)$, where $a$ is as in (10). Furthermore,*

    1. *If $\alpha$ is of Type 1, then*

$$\|\Delta_n\| = \mathcal{O}\left(\max\{n^{-\lambda\alpha_0}; n^{-\frac{a}{2}}; n^{-1.5}\}(\ln n)^{\frac{a}{2}+1}\right) \quad \textit{a.s. on } \mathcal{E}(x_*).$$

    2. *If $\alpha$ is of Type $\gamma$ with $2/b < \gamma < 1$, then*

$$\|\Delta_n\| = \mathcal{O}\left(\alpha_n(\alpha_n \ln t_{n+1})^{1/2} + (\alpha_n \ln t_{n+1})^{a/2}\right) \quad \textit{a.s. on } \mathcal{E}(x_*).$$

We refer the reader to the Appendix for the proofs of Lemmas 4.4 and 4.5. However, there is one point which we would like to emphasize here. That is, $\psi_n$ is a sum of scaled (matrix-valued) martingale differences. And, to derive its decay rate, we use the following law of iterated logarithm. Let $\mathrm{LL}(x) = \ln\ln(x)$.

**Lemma 4.6.** *For $n \geq 0$, let $U_{n+1} = \sum_{k=0}^{n} \phi_k \epsilon_{k+1}$, where $\{\epsilon_n\}$ is a real-valued martingale differ-ence sequence adapted to a filtration $\{\mathcal{F}_n\}$, and $\{\phi_n\}$ is a sequence of real-valued scalars, again adapted to $\{\mathcal{F}_n\}$.*

*Let $\{T_n\}$, also adapted to $\{\mathcal{F}_n\}$, be such that, for $n \geq 0$, $|\phi_n| \leq T_n$ a.s. and $\tau_n := \sum_{k=0}^{n} T_k^2$ satisfies $\lim_{n \to \infty} \tau_n = \infty$ a.s. Further, assume $\sup_{n \geq 0} \mathbb{E}[\epsilon_{n+1}^2 | \mathcal{F}_n] \leq \sigma^2$ a.s. for some constant $\sigma^2$. Also, let $\beta > 0$ be such that $\sum_n T_n^{2+2\beta} \tau_n^{-1-\beta} [\text{LL}(\tau_n)]^\beta < \infty$ and $\sup_{n \geq 0} \mathbb{E}\left[|\epsilon_{n+1}|^{2+2\beta} | \mathcal{F}_n\right] < \infty$ a.s. Then,*

$$\limsup [2\tau_n \text{LL}\tau_n]^{-1/2} |U_{n+1}| \leq \sigma \quad \text{a.s.} \tag{21}$$

**Remark 4.7.** *The condition $\sum_n T_n^{2+2\beta} \tau_n^{-1-\beta} [\text{LL}(\tau_n)]^\beta < \infty$ differs from the one in [12, Corol-lary 6.4.25]; in that, it includes the additional term $[\text{LL}(\tau_n)]^\beta$. The impact of this is that we no longer require $\beta$ to be in $(0,1)$ as was the case in [12, Corollary 6.4.25]. Instead, $\beta$ can now take any positive value. This is precisely what allows Theorem 2.1 to be applicable even when the stepsizes are non-square summable.*

**Remark 4.8.** *The above result goes through even if we have $\limsup_{n \to \infty} \mathbb{E}[\epsilon_{n+1}^2 | \mathcal{F}_n] \leq \sigma^2$ instead of $\sup_{n \geq 0} \mathbb{E}[\epsilon_{n+1}^2 | \mathcal{F}_n] \leq \sigma^2$; cf. [30, Result 1].*

We now present the proof of 4.1 which is a direct consequence of Lemmas 4.4 and 4.5.

*Proof of Lemma 4.1.* From (18), observe that

$$\|\mathbf{1}'\pi(x_n - x_*)\| \leq \|\psi_n\| + \|\Delta_n\|.$$

First consider the case where $\alpha$ is of Type $\gamma$. From Lemmas 4.4 and 4.5, we have

$$\limsup_{n \to \infty} \frac{\|\mathbf{1}'\pi(x_n - x_*)\|}{(\alpha_n \ln t_{n+1})^{1/2}} \leq C + \limsup_{n \to \infty} \mathcal{O}\left(\alpha_n + (\alpha_n \ln t_{n+1})^{(a-1)/2}\right) = C,$$

where the last display holds because $\lim_{n \to \infty} \alpha_n = 0$, $\lim_{n \to \infty} \alpha_n \ln t_{n+1} = 0$, and $a > 1$.

Next consider the case where $\alpha$ is of Type 1. Again, from Lemmas 4.4 and 4.5, we get

$$\limsup_{n \to \infty} \frac{\|\mathbf{1}'\pi(x_n - x_*)\|}{(\alpha_n \ln t_{n+1})^{1/2}} \leq C + \limsup_{n \to \infty} \mathcal{O}\left(\frac{\max(n^{-\lambda\alpha_0}; n^{-\frac{a}{2}}; n^{-1.5})(\ln n)^{1+a/2}}{(n^{-1}\ln n)^{1/2}}\right) = C,$$

where the last display holds because $\lambda\alpha_0 > \frac{1}{2}$ and $a > 1$.

The desired result now follows. $\qquad\square$

## 4.2 Bound on disagreement error $\|Qx_n\|$

We now turn to the detailed analysis of the disagreement component of the error. Let

$$\chi_{n+1} := \sum_{j=0}^{n} \alpha_j W^{n-j} Q M_{j+1} e^{-(t_{n+1}-t_{j+1})A}, \quad n \geq -1, \tag{22}$$

and

$$\Gamma_n := Qx_n - \chi_n, \quad n \geq 0. \tag{23}$$

Note that $\chi_n$ represents the cumulative noise in $Qx_n$. It is also easy to see that

$$\chi_{n+1} = W\chi_n e^{-\alpha_n A} + \alpha_n Q M_{n+1}, \quad n \geq 0. \tag{24}$$

We now state our bounds for $\|\chi_n\|$ and $\|\Gamma_n\|$. Note that $\chi_n$ and $\Gamma_n$ are peculiar to the DSA setup and do not have analogues in the one-timescale analysis.

**Lemma 4.9.** *Let $\delta > 0$. Then,*

$$\|\chi_{n+1}\| = \mathcal{O}\left(\alpha_n (\ln n)^{1+\delta}\right) \quad \text{a.s. on } \mathcal{E}(x_*) \tag{25}$$

**Lemma 4.10.** *Almost surely on $\mathcal{E}(x_*)$,*

$$\|\Gamma_{n+1}\| = \mathcal{O}\left(\alpha_n\right). \tag{26}$$

We refer readers to the Appendix for proofs of Lemma 4.9 and 4.10. The overall disagreement error can now be bounded as shown below.

*Proof of Lemma 4.2.* Observe that

$$\|Qx_n\| \le \|\Gamma_n\| + \|\chi_n\|.$$

Using lemmas 4.9 and 4.10, we then have

$$\|Qx_n\| = \mathcal{O}\left(\alpha_n\right) + \mathcal{O}\left(\alpha_n(\ln n)^{1+\delta}\right) = \mathcal{O}\left(\alpha_n(\ln n)^{1+\delta}\right),$$

as desired. □

## 5 Discussion

We derive a novel law of iterated logarithm for a family of nonlinear DSA algorithms that is useful in MARL. This law can also be seen as an asymptotic a.s. convergence rate result. It is the first of its kind in the distributed setup and holds under significantly weaker assumptions. Our proof uses a novel error decomposition and a novel law of iterated logarithm for a sum of martingale differences.

While our DSA framework is fairly general, a key limitation is that the scaling matrix (i.e., $A$) in each component function $h_i$ needs to be the same. It would be interesting to see if our approach can be extended to cover the general case [48] where the scaling matrices also depend on $i$. Another intriguing future direction is the setting with dynamic communication protocols, wherein the gossip matrix also evolves with time [10, 11]. A third direction is that of two-timescale DSA schemes [8, 6]. On the MARL side, important algorithms like distributed Q-learning [17] and its variants need more careful analysis and we believe our techniques would be instrumental for this as well. Finally, we would like to study the effect of momentum in MARL algorithms [1].

## Acknowledgments and Disclosure of Funding

We would like to thank Prof. Vivek Borkar for suggesting this exciting problem. We would also like to thank the anonymous reviewers for providing helpful and constructive feedback on the paper. Research of Gugan Thoppe is supported by IISc's start up grants SG/MHRD-19-0054 and SR/MHRD-19-0040.

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
