# 6  Appendix - Proofs

Throughout the appendix, we will presume that $\mathbb{P}\{\mathcal{E}(x_*)\} = 1$. This is mainly to avoid writing "a.s. on $\mathcal{E}(x_*)$" in every statement.

## 6.1  Agreement Error Results: Proof of Lemma 4.4

The discussion here builds upon the ideas used in the proof of [30, Lemma 1]. In order to not repeat everything, our focus below will only be on those arguments that differ from the original ones. Our first goal is to derive a relation that is similar to [30, (22)].

Let $\mu$ be an eigenvalue of $A$ with multiplicity $\nu$ and let $w'$ be a column vector in the null space of $(A - \mu I)^\nu$. That is, let $w'$ be a generalized right eigenvector of $A$ corresponding to the eigenvalue $\mu$. It is possible that both $\mu$ and $w$ are complex valued. Then, for any $t > 0$,

$$\mathrm{e}^{-tA}w' = \mathrm{e}^{-t(A-\mu\mathbb{I})}\mathrm{e}^{-t\mu\mathbb{I}}w' = \mathrm{e}^{-t\mu}\mathrm{e}^{-t(A-\mu\mathbb{I})}w' = \mathrm{e}^{-t\mu}\sum_{p=0}^{\nu-1}(-1)^p t^p w'_p,$$

where $w'_p = (A - \mu\mathbb{I})^p w'/p!$. Consequently,

$$\begin{aligned}
\psi_{n+1}w' &= \sum_{k=0}^{n}\alpha_k \mathbf{1}'\pi M_{k+1}\mathrm{e}^{-\mu(t_{n+1}-t_{k+1})}\sum_{p=0}^{\nu-1}(-1)^p(t_{n+1}-t_{k+1})^p w'_p \\
&= \mathbf{1}'\mathrm{e}^{-\mu t_{n+1}}\sum_{p=0}^{\nu-1}(-1)^p\sum_{k=0}^{n}\mathrm{e}^{\mu t_{k+1}}\alpha_k(t_{n+1}-t_{k+1})^p\pi M_{k+1}w'_p. \quad\quad (27)
\end{aligned}$$

Let $u$ be an arbitrary vector in $\mathbb{C}^d$ and, for $0 \le p \le \nu - 1$, set

$$G_{n+1}^{(p)} := \sum_{k=0}^{n}\mathrm{e}^{\mu t_{k+1}}\alpha_k(t_{n+1}-t_{k+1})^p\pi M_{k+1}u'.$$

Then, observe that

$$G_{k+1}^{(0)} - G_k^{(0)} = \mathrm{e}^{\mu t_{k+1}}\alpha_k\pi M_{k+1}u'.$$

Hence, for $p \ge 1$,

$$\begin{aligned}
G_{n+1}^{(p)} &= \sum_{k=0}^{n}(t_{n+1}-t_{k+1})^p(G_{k+1}^{(0)} - G_k^{(0)}) \\
&= \sum_{k=1}^{n}\left[(t_{n+1}-t_k)^p - (t_{n+1}-t_{k+1})^p\right]G_k^{(0)},
\end{aligned}$$

where the last relation follows since $G_0^{(0)} = 0$. Therefore, by the mean value theorem,

$$|G_{n+1}^{(p)}| \le p\sum_{k=1}^{n}\alpha_k(t_{n+1}-t_k)^{p-1}|G_k^{(0)}|.$$

We now use Lemma 4.6 to derive an almost sure upper bound of $|G_{n+1}^{(0)}|$. We will use this later to derive an almost sure bound on $|G_{n+1}^{(p)}|$ for $1 \le p \le \nu - 1$.

By applying individually to the real and imaginary parts, it is not difficult to see that Lemma 4.6 is true even if the terms $\epsilon_k$ and $\phi_k$ in its statement are complex numbers. Keeping this in mind, let $\phi_k = \mathrm{e}^{\mu t_{k+1}}\alpha_k$ and $\epsilon_{k+1} = \pi M_{k+1}u'$.

We now verify the assumptions of Lemma 4.6. Clearly, $\phi_k$ and $\epsilon_{k+1}$ are complex scalars. Also, $\{\epsilon_{k+1}\}$ is a martingale difference sequence. Pick a $\beta > 0$ such that $1/\gamma < 1 + \beta < b/2$; this is possible since $b$ in $\mathcal{A}_4$ is bigger than 2 and $\gamma > 2/b$. Then, because of $\mathcal{A}_4$ and the fact that

$$\mathbb{E}[|\epsilon_{n+1}|^{2+2\beta}|\mathcal{F}_n] \le \|u\|^{2+2\beta}\,\mathbb{E}[\|\pi M_{n+1}\|^{2+2\beta}|\mathcal{F}_n],$$

we have $\sup_{n\geq 0} \mathbb{E}[|\epsilon_{n+1}|^{2+2\beta}|\mathcal{F}_n] < \infty$ a.s. Also, $\mathbb{E}[|\epsilon_{n+1}|^2|\mathcal{F}_n] = u\mathbb{E}[M'_{n+1}\pi'\pi M_{n+1}|\mathcal{F}_n]u'$. Combining this with $\mathcal{A}_4$, it then follows that $\limsup_{n\to\infty} \mathbb{E}[|\epsilon_{n+1}|^2|\mathcal{F}_n] = uMu'$ a.s.

Let $\lambda = \mathscr{R}(\mu)$. Since $-A$ is Hurwitz, we have $\lambda \geq \lambda_{\min} > 0$. Hence, if $T_n = \alpha_n e^{\lambda t_{n+1}}$, then $|\phi_n| \leq T_n$. Further, if $\alpha$ is of Type 1, then clearly $\tau_n \sim [\alpha_0/(2\lambda\alpha_0 - 1)]\alpha_n e^{2\lambda t_{n+1}}$; combining this with the fact that $\alpha_0 \geq 1/(2\lambda_{\min})$, then shows $\sum_{n\geq 0} T_n^{2+2\beta} \tau_n^{-1-\beta}[\text{LL}(\tau_n)]^\beta \sim C\sum_{n\geq 0}[\text{LL}(n)]^\beta/n^{1+\beta} < \infty$. On the other hand, if $\alpha$ is of Type $\gamma$, then [30, Lemma 4] shows that $\tau_n \sim 1/(2\lambda)\alpha_n e^{2\lambda t_{n+1}}$; this along with the fact that $\gamma(1+\beta) > 1$ then shows that $\sum_{n\geq 0} T_n^{2+2\beta} \tau_n^{-1-\beta}[\text{LL}(\tau_n)]^\beta \sim (2\lambda)^{1+\beta}\sum_{n\geq 0}[\ln(n)]^\beta/n^{\gamma(1+\beta)} < \infty$. This verifies all the conditions needed in Lemma 4.6.

It now follows from Lemma 4.6 that, almost surely,

$$\limsup_{n\to\infty} \frac{e^{-\lambda t_{n+1}}|G_{n+1}^{(0)}|}{[\alpha_n \ln t_{n+1}]^{1/2}} \leq \sqrt{\frac{uMu'}{\lambda - \zeta}},$$

where

$$\zeta = \begin{cases} 1/(2\alpha_0), & \text{if } \alpha \text{ is Type 1,} \\ 0, & \text{if } \alpha \text{ is Type } \gamma. \end{cases}$$

This expression is exactly of the form given in (22) in [30]; therefore, by repeating the arguments that follow (22) there, we get

$$\limsup_{n\to\infty} \frac{e^{-\lambda t_{n+1}}|G_{n+1}^{(p)}|}{[\alpha_n \ln t_{n+1}]^{1/2}} \leq \Lambda_p$$

for some deterministic constant $\Lambda_p$. Combining this with (27) then gives

$$\limsup_{n\to\infty} \frac{\|\psi_{n+1}w'\|}{[\alpha_n \ln t_{n+1}]^{1/2}} \leq \sqrt{m}\Lambda_w$$

for some deterministic constant $\Lambda_w$; $\sqrt{m}$ comes in the expression due to the vector $\mathbf{1}'$.

Because $w$ was arbitrary, the desired result now follows.

## 6.2 Agreement Error Results: Proof of Lemma 4.5

Using (17), observe that

$$\psi_{n+1} = \psi_n e^{-(t_{n+1}-t_n)A} + \alpha_n \mathbf{1}'\pi M_{n+1},$$

which, using a version of Taylor's theorem for matrix valued functions, can be written as

$$\psi_{n+1} = \psi_n[\mathbb{I} - \alpha_n A + O(\alpha_n^2)\mathbb{I}] + \alpha_n \mathbf{1}'\pi M_{n+1}.$$

Separately, recall from (18) that

$$\Delta_{n+1} = \mathbf{1}'\pi(x_{n+1} - x_*) - \psi_{n+1}.$$

Using (2), (6), (8), and $\mathcal{A}_2$, it then follows that

$$\Delta_{n+1} = \Delta_n(\mathbb{I} - \alpha_n A) + \alpha_n \mathbf{1}'\pi f_1(x_n) + \psi_n O(\alpha_n^2).$$

For $r_{n+1} := \mathbf{1}'\pi f_1(x_n)$, we finally have

$$\Delta_{n+1} = \Delta_n[\mathbb{I} - \alpha_n A] + \alpha_n[r_{n+1} + O(\alpha_n)\psi_n].$$

Next, let $0 < \lambda < \hat{\lambda} < \lambda_{\min}$. Then, for all sufficiently large $n$, $\|\mathbb{I} - \alpha_n A\| \leq (1 - \hat{\lambda}\alpha_n)$; therefore, for some $C \geq 0$,

$$\|\Delta_{n+1}\| \leq (1 - \alpha_n\hat{\lambda})\|\Delta_n\| + C\alpha_n(\|r_{n+1}\| + \alpha_n\|\psi_n\|),$$

$$\leq (1 - \alpha_n \hat{\lambda}) \|\Delta_n\| + C\alpha_n (\alpha_n \|\psi_n\| + \|\psi_n\|^a + \|\Delta_n\|^a),$$

where the last relation follows by using (10) and (18).

Equivalently, for all large enough $n$,

$$\|\Delta_{n+1}\| \leq (1 - \lambda\alpha_n)\|\Delta_n\| - \alpha_n[(\hat{\lambda} - \lambda) - C\|\Delta_n\|^{a-1}]\|\Delta_n\| + C\alpha_n(\alpha_n\|\psi_n\| + \|\psi_n\|^a).$$

Since $\|\psi_n\| \to 0$ and $x_n \to x_*$ a.s., we have $\|\Delta_n\| \to 0$ a.s. This, along with the fact that $a > 1$, then shows that $(\hat{\lambda} - \lambda) - \|\Delta_n\|^{a-1} \geq 0$ for all large enough $n$. Hence, for all large enough $n$,

$$\|\Delta_{n+1}\| \leq (1 - \lambda\alpha_n)\|\Delta_n\| + C\alpha_n[\alpha_n\|\psi_n\| + \|\psi_n\|^a].$$

Thus,

$$\|\Delta_n\| = \mathcal{O}\left(\max\left\{e^{-\lambda\sum_{k=0}^n \alpha_k}, \sum_{j=0}^n \alpha_j e^{-\lambda\sum_{k=j+1}^n \alpha_k}[\alpha_j\|\psi_j\| + \|\psi_j\|^a]\right\}\right),$$

as desired in (20).

To proceed with further calculations, let

$$\rho_n := \sum_{j=0}^n \alpha_j e^{-\lambda(t_{n+1} - t_{j+1})}(\alpha_j\|\psi_j\| + \|\psi_j\|^a).$$

When $\alpha$ is of Type 1, $e^{-\lambda t_{n+1}} = \Theta(n^{-\lambda\alpha_0})$. This, combined with Lemma 4.4, then shows that

$$\rho_n = \mathcal{O}\left(n^{-\lambda\alpha_0}\sum_{j=0}^n \frac{j^{\lambda\alpha_0}}{j}\left[\frac{1}{j}\left(\frac{\ln j}{j}\right)^{\frac{1}{2}} + \left(\frac{\ln j}{j}\right)^{\frac{a}{2}}\right]\right)$$

$$= \mathcal{O}\left(n^{-\lambda\alpha_0}\sum_{j=0}^n \left[j^{\lambda\alpha_0 - 2.5}(\ln j)^{1/2} + j^{\lambda\alpha_0 - 1 - \frac{a}{2}}(\ln j)^{\frac{a}{2}}\right]\right)$$

$$= \mathcal{O}\left(n^{-\lambda\alpha_0}\left[n^{\lambda\alpha_0 - 1.5}(\ln n)^{1/2} + n^{\lambda\alpha_0 - \frac{a}{2}}(\ln n)^{\frac{a}{2}} + (\ln n)^{\frac{a}{2}+1}\right]\right)$$

$$= \mathcal{O}\left(n^{-1.5}(\ln n)^{1/2} + n^{-\frac{a}{2}}(\ln n)^{\frac{a}{2}} + n^{-\lambda\alpha_0}(\ln n)^{\frac{a}{2}+1}\right) \quad \text{a.s.};$$

the $\ln n$ factor in the last term of the third relation accounts for the possibility of $\lambda\alpha_0$ being either 2.5 or $1 + a/2$.

On the other hand, when $\alpha$ is of Type $\gamma$ with $2/b < \gamma < 1$, Lemma 4.4 shows that

$$\rho_n = \mathcal{O}\left(e^{-\lambda t_{n+1}}\sum_{j=1}^n \alpha_j e^{\lambda t_{j+1}}\left[\alpha_j(\alpha_j \ln t_{j+1})^{\frac{1}{2}} + (\alpha_j \ln t_{j+1})^{\frac{a}{2}}\right]\right).$$

It then follows from [30, Lemma 4] that

$$\rho_n = \mathcal{O}\left(\alpha_n(\alpha_n \ln t_{n+1})^{\frac{1}{2}} + (\alpha_n \ln t_{n+1})^{\frac{a}{2}}\right) \quad \text{a.s.}$$

The desired result is now easy to see.

## 6.3 Disagreement Error Results: Proof of Lemma 4.9

Because of $\mathcal{A}_1$, note that all eigenvalues of $W$ have magnitude less than or equal to 1. Furthermore, there is one and only one eigenvalue with magnitude 1 and that eigenvalue is 1 itself. Recall from (6) that the left eigenvector of $W$ corresponding to the eigenvalue 1 is $\pi$.

For ease of exposition, we will presume that every other eigenvalue of $W$ has multiplicity 1. Similarly, we will presume that every eigenvalue of $A$ has multiplicity 1. The general case where some of the eigenvalues may have multiplicities larger than 1 can be handled using the ideas from this proof along with those used in the proof of Lemma 4.4 (or [30, Lemma 1]) and Lemma 6.1.

Recall from (22) that

$$\chi_{n+1} = \sum_{j=0}^{n} \alpha_j W^{n-j} Q M_{j+1} e^{-(t_{n+1}-t_{j+1})A}.$$

To prove the desired result, it suffices to show there exists some constant $C \geq 0$ such that

$$\limsup_{n \to \infty} \frac{|u\chi_{n+1}v'|}{\alpha_n (\ln n)^{1+\delta}} \leq C \quad \text{a.s.} \tag{28}$$

for arbitrary row vectors $u \in \mathbb{R}^m$ and $v \in \mathbb{R}^d$, both with unit norm. Now, due to the above assumption on multiplicities, the eigenvalues of $W$ span $\mathbb{R}^m$, while the eigenvalues of $A$ span $\mathbb{R}^d$. Hence, it suffices to show (28) when $u$ is a left eigenvector of $W$ and $v'$ is a right eigenvector of $A$.

When $u = \pi$, we have $u\chi_{n+1}v' = 0$. This follows from (6) and the fact that $\pi Q = 0$. The desired result thus trivially holds in this case. Keeping this in mind, suppose that $u$ is some left eigenvector of $W$ that is not equal to $\pi$. We will presume that the eigenvalue corresponding to $u$ is $re^{\iota\theta}$, where $\iota := \sqrt{-1}$. Due to $\mathcal{A}_1$, $r < 1$; on the other hand, $\theta \in [0, 2pi)$ can be arbitrary. Similarly, let $\lambda + \iota\sigma$ denote the eigenvalue of $A$ corresponding to $v'$.

Now observe that

$$u\chi_{n+1}v' = \sum_{j=0}^{n} \alpha_j u W^{n-j} Q M_{j+1} e^{-(t_{n+1}-t_{j+1})A} v',$$

$$= \sum_{j=0}^{n} \alpha_j r^{n-j} e^{\iota\theta(n-j)} e^{-(t_{n+1}-t_{j+1})(\lambda+\iota\sigma)} u Q M_{j+1} v'$$

$$= r^n e^{\iota\theta n} e^{-(\lambda+\iota\sigma)t_{n+1}} Z_{n+1},$$

where

$$Z_{n+1} := \sum_{j=0}^{n} \alpha_j r^{-j} e^{-j\iota\theta} e^{(\lambda+\iota\sigma)t_{j+1}} u Q M_{j+1} v'.$$

Hence,

$$|u\chi_{n+1}v'| = r^n e^{-\lambda t_{n+1}} |Z_{n+1}|. \tag{29}$$

Note that $\{Z_{n+1}\}$ is one-dimensional martingale sequence, possibly complex valued. Hence, to derive a bound on $|Z_{n+1}|$, we now make use of [12, Theorem 6.4.24]. The result in [12] is stated for real-valued martingale sequences. To account for this discrepancy, we separately deal with the real and imaginary parts of $Z_{n+1}$.

Let $R_{n+1} := \mathscr{R}(Z_{n+1})$ and $I_{n+1} := \Im(Z_{n+1})$ denote the real and imaginary parts of $Z_{n+1}$, respectively. Also, for $n \geq 0$, let

$$L_n := \sup_{0 \leq j \leq n} \mathbb{E}[|uQM_{j+1}v'|^2 | \mathcal{F}_j].$$

Due to $\mathcal{A}_4$.b and the fact that $x_n \to x_*$ a.s., note that

$$\sup_{n \geq 0} L_n < \infty \quad a.s. \tag{30}$$

Then, it is not difficult to see that the quadratic variation of $(R_n)$ satisfies

$$\langle R \rangle_{n+1} \leq C \sum_{k=0}^{n} \alpha_k^2 r^{-2k} e^{2\lambda t_{k+1}} \mathbb{E}[|uQM_{k+1}v'|^2 | \mathcal{F}_k]$$

$$\leq CL_n \sum_{k=0}^{n} \alpha_k^2 r^{-2k} e^{2\lambda t_{j+1}}$$

$$\leq CL_n \alpha_n^2 r^{-2n} e^{2\lambda t_{n+1}}.$$

Let $K_n = CL_n$, $n \geq 0$, where $C$ is the constant present in the last relation above.

Next, define $s_n = \sqrt{K_n} \alpha_n r^{-n} e^{\lambda t_{n+1}} (\ln n)^{1/2+\delta}$ for some $\delta > 0$. Then, it is easy to see that $s_n \to \infty$ and $\langle R \rangle_{n+1} \leq s_n^2$. Further,

$$|R_{n+1} - R_n| \leq \alpha_n r^{-n} e^{\lambda t_{n+1}} |uQM_{n+1}v'|$$

$$\leq C\alpha_n r^{-n} e^{\lambda t_{n+1}} [1 + ||Q(x_n - x^*)||],$$

where the last relation follows due to $\mathcal{A}_4$.b and the fact that both $||u||$ and $||v||$ are bounded from above by $1$. Let $\hat{K}$ be the constant in the last relation above.

Then, for $h(x) = \sqrt{2x \ln \ln x}$, we have

$$|R_{n+1} - R_n| \leq \frac{\sqrt{2}\hat{K}}{\sqrt{K_n}} \frac{s_n^2}{h(s_n^2)} \frac{[1 + ||Q(x_n - x^*)||]}{(\ln n)^{1/2+\delta}} (\ln \ln s_n^2)^{1/2}$$

$$= \frac{C_n s_n^2}{h(s_n^2)},$$

where

$$C_n = \frac{\sqrt{2}\hat{K}}{\sqrt{K_n}} \frac{[1 + ||Q(x_n - x^*)||]}{(\ln n)^{1/2+\delta}} (\ln \ln(s_n^2))^{1/2}.$$

Since $x_n \to x^*$ a.s., $||Q(x_n - x^*)||$ is bounded from above a.s. and, hence, $C_n \to 0$ a.s. Applying [12, Theorem 6.4.24], it now follows that

$$\limsup_{n \to \infty} \frac{|R_{n+1}|}{h(s_n^2)} \leq 1 \quad \text{a.s.}$$

Similarly, it can be shown that

$$\limsup_{n \to \infty} \frac{|I_{n+1}|}{h(s_n^2)} \leq 1 \quad \text{a.s.}$$

Combining the two relations above then shows that

$$\limsup_{n \to \infty} \frac{|Z_{n+1}|}{h(s_n^2)} \leq \sqrt{2} \quad \text{a.s.}$$

By substituting this in (29) and then making use of (30), we finally get

$$\limsup_{n \to \infty} \frac{|u\chi_{n+1}v'|}{\alpha_n (\ln n)^{1+\delta}} \leq C \quad \text{a.s.}$$

This verifies (28), as desired.

## 6.4 Disagreement Error Results: Proof of Lemma 4.10

The operator $Q = \mathbb{I} - \mathbf{1}'\pi$ satisfies the simple properties $WQ = QW$ and $Q\mathbf{1}'\pi = 0$. These properties and $\mathcal{A}_2$ lend (2) into

$$Qx_{n+1} = WQx_n + \alpha_n(Qh(x_n) + QM_{n+1}),$$
$$= WQx_n + \alpha_n(Q(B + f_2(x_n)) + QM_{n+1}).$$

Using this and the definition of $\Gamma_n$ from (23) then shows that

$$\Gamma_{n+1} = Qx_{n+1} - \chi_{n+1}$$
$$= W\Gamma_n + W\chi_n\kappa_n + \alpha_n Q(B + f_2(x_n)),$$

where $\kappa_n = \mathbb{I} - e^{-\alpha_n A}$. Since $\Gamma_0 = Qx_0$, by unrolling the previous relation, we get

$$\Gamma_{n+1} = W^{n+1}Qx_0 + \sum_{j=0}^{n} \alpha_j W^{n-j} Q(B + f_2(x_j)) + \sum_{j=0}^{n} W^{n+1-j} \chi_j \kappa_j.$$

For ease of discussion, we will presume that $W$ has unique eigenvalues. The general case where some of the eigenvalues may have multiplicities larger than 1 can be handled by building upon the ideas discussed in the proof of Lemma 4.4 (or [30, Lemma 1]) and Lemma 6.1.

Using (22), (6), and the fact that $\pi Q = 0$, it is easy to see that $\pi\Gamma_{n+1} = 0$. Hence, the desired result trivially holds then. Now, let the row vector $u \neq \pi$, of unit norm, be an arbitrary left eigenvector of $W$ and suppose that its eigenvalue is $\rho e^{\iota\theta}$. Because of $\mathcal{A}_1$, it must be the case that $\rho < 1$.

It is then easy to see that

$$u\Gamma_{n+1} = \rho^{n+1} e^{\iota(n+1)\theta} uQx_0 + \sum_{j=0}^{n} \alpha_j \rho^{n-j} e^{\iota(n-j)\theta} uQ(B+f_2(x_n)) + \sum_{j=0}^{n} \rho^{n+1-j} e^{\iota(n+1-j)\theta} u\chi_j \kappa_j.$$

Now, since $x_n \to x_*$ a.s., it follows that $x_n$ is bounded a.s. Hence,

$$\|u\Gamma_{n+1}\| \leq C\rho^{n+1} + C\sum_{j=0}^{n} \alpha_j \rho^{n-j} + C\sum_{j=0}^{n} \rho^{n+1-j} \|\chi_j\|\|\kappa_j\| \quad \text{a.s.}$$

From Lemma 4.9, $\|\chi_j\| = O(\alpha_j(\ln j)^{1+\delta})$ a.s. Separately, $\|\kappa_j\| = O(\alpha_j)$. Hence,

$$\|u\Gamma_{n+1}\| \leq C\rho^{n+1} + C\sum_{j=0}^{n} \alpha_j \rho^{n-j} + C\sum_{j=0}^{n} \alpha_j^2 \rho^{n-j}(\ln j)^{1+\delta} \quad \text{a.s.}$$

Next, note that $\alpha_j^{0.5} \ln j = o(1)$ for either type of the step sizes. Additionally, between $\sum_{j=0}^{n} \alpha_j \rho^{n-j}$ and $\sum_{j=0}^{n} \alpha_j^{1.5} \rho^{n-j}$, the dominant term is the former; this is because each term in its summation dominates the corresponding term in the latter. Separately,

$$\sum_{j=0}^{n} \alpha_j \rho^{n-j} \leq \left(\max_{0 \leq j \leq n} \rho^{(n-j)/2}\alpha_j\right) \sum_{j=0}^{n} \rho^{(n-j)/2} = O(\alpha_n).$$

The desired result is now easy to see.

## 6.5  Proof of Auxiliary Lemma 4.6

We only give a sketch of the proof since the arguments are similar to the ones used in the derivation[7] of [12, Corollary 6.4.25]. In the latter's proof, a sequence $\{C_n\}$ adapted to $\{\mathcal{F}_n\}$ needs to be chosen such that

$$\lim_{n\to\infty} C_n = 0 \quad \text{and} \quad \sum_{n\geq 0} \frac{|\phi_n|^2 T_n^{2\beta} \mathrm{LL}(s_n^2)^{\beta-1}}{C_n^{2\beta} s_n^{2+2\beta}} < \infty,$$

where $s_n^2 = \sigma^2 \tau_n$.

In [12], $C_n^2$ was chosen to be $[\mathrm{LL}(s_n^2)]^{1-1/\beta}$. Because we need $\lim_n C_n$ to be 0, it necessarily follows that $\beta$ should be in $(0, 1)$.

In contrast, we set $C_n^2 = [\mathrm{LL}(s_n^2)]^{-1/\beta}$. Since $\tau_n \to \infty$, we have that $s_n \to \infty$ as well. Combining this with the fact that $\beta > 0$, it then follows that $C_n \to 0$, as desired. Separately, due to the given conditions,

$$\sum_{n\geq 0} \frac{|\phi_n|^2 T_n^{2\beta} \mathrm{LL}(s_n^2)^{\beta-1}}{C_n^{2\beta} s_n^{2+2\beta}} \leq \sum_{n\geq 0} \frac{T_n^{2+2\beta}[\mathrm{LL}(s_n^2)]^{\beta-1}}{C_n^{2\beta} s_n^{2+2\beta}} = \frac{1}{\sigma^{2+2\beta}} \sum_{n\geq 0} T_n^{2+2\beta} \tau_n^{-(1+\beta)}[\mathrm{LL}(s_n^2)]^{\beta} < \infty.$$

The desired result now follows.

---

[7]The definition of $\xi_{k+1}$ and the expression for $N_{n+1}$, as given in [12, p212], has typos. The correct versions are $\xi_{k+1} := \frac{C_k^\beta s_k^{2\beta}}{T_k^\beta [h(s_k^2)]^\beta} \eta_{k+1} \mathbb{1}[T_k \neq 0]$ and $N_{n+1} = \sum_{k=0}^{n} \frac{\phi_k T_k^\beta [h(s_k^2)]^\beta}{C_k^\beta s_k^{2\beta} h(s_k^2)} \xi_{k+1}$

## 6.6 Auxiliary Linear Algebraic Lemma

We state a linear algebra result here that is useful in the proof of our results when some of the eigenvalues of $W$ have multiplicities bigger than $1$.

**Lemma 6.1.** *Suppose that the Jordan normal form of $W$ has $L$ Jordan blocks with sizes $\ell_1, \ldots, \ell_L$, respectively. Also, let $1 = \theta_0, \theta_1, \ldots, \theta_{L-1}$ denote the corresponding distinct eigenvalues. Then, there exist some matrices $J_{i\ell}$, $1 \leq i \leq L - 1$ and $0 \leq \ell \leq \ell_i - 1$, such that, for $k \geq 0$,*

$$W^k Q = \sum_{i=1}^{L-1} \sum_{\ell=0}^{\ell_i-1} \theta_i^{k-\ell} \binom{k}{\ell} J_{i\ell}. \tag{31}$$

*Proof.* Due to $\mathcal{A}_1$, recall that precisely one eigenvalue of $W$ equals $1$. And, the left and right eigenvectors of $W$ corresponding to this eigenvalue are $\pi$ and $\mathbf{1}$, respectively. Therefore, a simple Jordan decomposition shows that

$$W^k = \sum_{i=1}^{L-1} \sum_{\ell=0}^{\ell_i-1} \theta_i^{k-\ell} \binom{k}{\ell} J_{i\ell} + \mathbf{1}\pi.$$

for some suitably defined matrices $\{J_{i\ell}\}$. Separately, since $W\mathbf{1} = \mathbf{1}$, we have $W^k Q = W^k(\mathbb{I} - \mathbf{1}\pi) = W^k - \mathbf{1}\pi$. The desired result is now straightforward to see. $\square$