# OpenReview forum: "A Law of Iterated Logarithm for Multi-Agent Reinforcement Learning"
_NeurIPS.cc/2021/Conference — NeurIPS 2021 Poster_

### Official Review · Reviewer_jy9o · 2021-07-16

**Rating:** 7
**Confidence:** 4

**Summary:**

This paper studies asymptotic convergence rates for a family of distributed Stochastic Approximation Schemes in a cooperative multi-agent learning setting (Nonlinear Gossip models). They provide convergence rates which applies almost surely on sample paths, improving over the existing analysis that analyse the expected behavior. In addition they have weaker assumptions on the Gossip matrix and the step sizes (removing square summability).

As a result, they analyze policy evaluation through TD(0) with linear function approximation for the infinite horizon reinforcement learning problem and the convergence rates follow for TD(0).



**Limitations And Societal Impact:**

See above for limitations.

**Main Review:**

The model studied here is not widely studied in ML literature, but they show how their results apply to TD(0) with linear function approximation in the MARL setting. The main improvement is providing convergence rates over sample paths, eventhough it is asymptotic. This applies to a subclass of cooperative multi-agent RL where agents use TD(0).

1) Could this framework be used to analyze a competitive setting or where the agents maybe negatively correlated? For instance, the setting of Markov games (Daskalakis et al. 2021)?

2) Are there any conditions that guarantee the uniqueness of $x^*$?

3) I believe a reason for the asymptotic nature is moving from the discrete updates to the limiting behavior of the ODE and I feel it would be better to add more discussion on this part (even if some of it comes from a different reference).

4) What are the challenges of understanding other function approximation schemes, maybe quadratic (for example), could this be subsumed in the function h(x)?

5) For the reduction to the RL setting the IID assumption seems to be key to ensure A and B dont depend on n, is there a work around for this? Would it be fruitful to look at the state-action visitation distribution?

Additional Reference:
Daskalakis, Constantinos, Dylan J. Foster, and Noah Golowich. "Independent policy gradient methods for competitive reinforcement learning." arXiv preprint arXiv:2101.04233 (2021).

**Time Spent Reviewing:**

7

---

> ### Author Response · Authors · 2021-08-10
> **Competitive MARL, Uniqueness of x_*, Function Approximation, and IID Assumption,**
>
> We thank the reviewer for posing interesting questions. Below, we answer them one by one.
>
> 1. Thanks for pointing out [Daskalakis et al. 2021]. It indeed looks to be an interesting paper to read. Our approach to analyze the problem discussed there would have been the following. For the sake of simplicity, suppose that there are no projection operators in (3) of [Daskalakis et al. 2021]. Since there is no explicit communication between the agents about their respective iterates, it follows that the gossip matrix here would be the identity matrix. Hence, it makes sense to view it as a two-timescale stochastic approximation algorithm rather than as a DSA. Subsequently, it would be useful to see if the results stated in [27] (which discusses a variant of our result in the two-timescale setup) are applicable in this setup or not.
>
> 2. One sufficient condition is that the limiting ODE given in (4) has a unique globally asymptotically stable equilibrium. Note that this is indeed the case for the distributed TD(0) algorithm with linear function approximation.
>
> 3. We apologize but we were unable to fully understand this question. The limiting ODE in (4) is only used to define a potential limit point for the DSA algorithm. As such, it does not play a direct role in any of our proofs. To obtain a finite time result in our distributed setup, a possible approach would be to build upon the ideas discussed in [6].
>
> 4. This is an interesting question. A quick calculation using quadratic function approximation shows that Assumption $A_2$ does not hold true in that case. Hence, one would have to rederive the entire proof for this new setup. We emphasize that this observation is not peculiar to our result in the distributed setup alone. In fact, a similar conclusion can be made even for centralized single-timescale stochastic approximation algorithms of the kind discussed in [33]. (see also our response to reviewer KYP4 on examples of nonlinear settings where our result would be applicable).
>
> 5. In general, without the IID assumption, $A$ and $B$ would have to be defined via the stationary distribution of the Markov chain induced by the behavior policy; see [4, Chapter 6.2]. Indeed, the state visitation frequency would converge to this distribution. Thus, even in that case $A$ and $B$ would not depend on n. However, without the IID assumption, the DSA algorithm would have an additional Markov noise, analysis of which is beyond the scope of this work.

---

> > ### Comment · Reviewer_jy9o · 2021-08-23
> > **Thank you for the response**
> >
> > Dear authors,
> >
> >  Thank you for the detailed response. After carefully reading through other reviews and the responses, I am happy to increase my score to Accept.
> >
> > Kindly add the discussion relating to $h(x)$ with the examples, comments on competitive MARL (Daskalakis et al. 2021) to enhance the quality of the paper.

---

> > > ### Author Response · Authors · 2021-08-24
> > > **Thank you for the positive response**
> > >
> > > Dear Reviewer,
> > >
> > > We thank you for your positive feedback. Indeed, we will add the discussion that you have mentioned.

---

### Official Review · Reviewer_Yt7X · 2021-07-18

**Rating:** 8
**Confidence:** 3

**Summary:**

This paper presents a Law of Iterated Logarithm (LIL) for Distributed Stochastic Approximation (DSA) schemes, which characterizes the rate of convergence of the updates to their stable equilibrium. The conditions are more general than previous works (in particular the gossip matrix is not required to be doubly stochastic and step-sizes need not be square summable). An application to the case of TD(0) with linear function approximation is given.

**Limitations And Societal Impact:**

As mentioned above, some further discussion about the implications of the LIL would be appreciated.

**Main Review:**

This is a well-written paper with a seemingly strong technical contribution to the DSA literature (although I am not an expert in that area), so I am recommending acceptance.

The writing is very clear, in particular the related works are well-covered. The proof outline (Section 4) is well sketched-out.

One question that I had was about the nature of the convergence guarantee. The theorem statement gives a convergence rate which holds “almost surely on E(x_\star)”, where E(x_\star) is the set of sequences which converges to x_star. It seems that this does not actually quantify the probability of convergence, but simply says that IF x_n converges to x_\star then the rate will be as given (and this rate holds with probability one). Am I interpreting this correctly? If so, are there other results in the literature guaranteeing convergence with high probability (or with probability one) for the same conditions considered here? Some discussion about this point in the text would be appreciated.

I am also wondering about the applicability of this theorem to other RL algorithms? The discussion mentions that Q-learning would require extending the analysis but have the authors tried applying this theorem to other policy evaluation algorithms (which are more likely to be “linear” and could fall under the umbrella of Assumption 2). Some discussion on this point in the text would also be helpful.

**Time Spent Reviewing:**

4

---

> ### Author Response · Authors · 2021-08-10
> **Nature of convergence guarantees and applicability in more general settings**
>
> $\textbf{Nature of convergence guarantees and results in literature discussing convergence guarantees}$
>
> We sincerely thank the reviewer for the positive feedback. Yes, the interpretation is correct: our result discusses convergence rates only along those sample paths where the DSA scheme converges to $x_*.$ Indeed, there already exist results in the literature that provide sufficient conditions for convergence of DSA algorithms. We refer the reviewer to [26] where one such result is discussed. The similarities and differences between the assumptions in [26] and ours are as follows:
>
> 1. Property of the gossip matrix $W:$ Both [26] and our work only require that the gossip matrix be row stochastic (and not doubly stochastic). However, [26] additionally allows the gossip matrix to also depend on the iteration index $n.$ As part of our future work, we plan to derive convergence rates to handle this case as well.
> 2. Assumption on $h:$ [26] only requires that h be Lipschitz continuous, whereas we assume some additional structure. We emphasize that our assumption is a natural generalization of assumption (A1) in [33], which also discusses a law of iterated logarithm for one-timescale stochastic approximation (see also our response to reviewer KYP4, where we have provided an intuitive explanation for $A_2$)
> 3. Assumptions on the step-size sequence $(\alpha_n)$: [26] requires that the stepsizes be square summable. In contrast, our result is also applicable to non-square summable stepsizes. Thus, our assumption is significantly weaker than its analogue in [26].
> 4. Assumption on the noise sequence $(M_n)$: Both [26] and our work requires that $(M_n)$ be a martingale-difference sequence. Additionally, while [26] only requires that $||M_{n  + 1}|| \leq C(1 + ||x_n||),$ $n \geq 0,$ for some constant $C \geq 0$ (see (A2’) in [26]), we require that $||Q M_{n + 1}|| \leq C(1 + ||x_n - x_*||)$ for some constant $C \geq 0.$ While our assumption does not follow directly from the assumption made in [26], as shown in Section 3 of our paper, we believe that our assumption should hold in almost all settings where (A2’) of [26] holds. Finally, we require two more assumptions on $(M_n).$ These are similar in spirit to those assumed in [33].
>
> \$\textbf{Applicability of the result in more general settings (e.g., Q-learning and other policy evaluation schemes)}$
>
> With regards to results on distributed Q-learning, we are only aware of [14]. In that, convergence in the tabular case is discussed. However, even that result only applies to a special case where the gossip matrix converges to the identity matrix asymptotically (see (10) and (11) in [14]). Hence, as a first step, it would be interesting to see if the convergence of distributed Q-learning itself holds for more general gossip matrices. Our guess is that results concerning stochastic approximation with fixed point iteration from [4, Chapter 10] should be of use here. By building upon this result and a novel concentration bound obtained recently in [Borkar 2021], we believe it should be possible to obtain convergence rates as well.
>
> From a practitioners’ perspective, especially when the state space is large, the more interesting and also challenging case is that of distributed Q-learning with function approximation. To the best of our knowledge, no convergence and convergence rate result exists in this case. For handling these questions, one would need some non-trivial effort. Here again, results from [4, Chapter 10] and [Borkar 2021] should be of use. There is, however, one case where it is possible that our proof ideas are directly useful. This is when the behavior policy is close to the optimal policy in the sense discussed in Assumption 3.3. of [Chen et al, 2020]. Our guess is that one should then be able to replace the max operator in the Q-learning algorithm with a simple average, thereby facilitating the application of analysis discussed in this work.
>
> On application of our results to other policy evaluation algorithms such as GTD(0), TDC, and GTD2, we wish to state the following. In their generic form, these algorithms have a two-timescale nature. Therefore, our results will not directly apply to them. However, there are some specific choices of stepsize sequences when these algorithms indeed become one timescale  and one can then ask if our result applies to their distributed variants. Since these schemes are linear in nature, we strongly believe the answer to this question must be a yes. We will rigorously check this in the review period. If it is indeed the case, we will definitely mention this in the paper.
>
> [Borkar 2021]: Borkar, Vivek S. "A concentration bound for contractive stochastic approximation." Systems & Control Letters 153 (2021): 104947.
>
> [Chen et al 2019]: Chen, Zaiwei, et al. "Finite-sample analysis of nonlinear stochastic approximation with applications in reinforcement learning." arXiv preprint arXiv:1905.11425 (2019).

---

### Official Review · Reviewer_ftij · 2021-07-19

**Rating:** 6
**Confidence:** 2

**Summary:**

The paper introduces a new distributed stochastic approximations scheme that can be exploited in the Multi-Objective RL (MARL). They show that for TD(0) with linear function approximation, we can enjoy the convergence rate of $\mathcal{O}(\sqrt{n^{-\gamma}\log n})$.

**Limitations And Societal Impact:**

The limitations and societal impact have been adequately addressed.

**Main Review:**


**Originality and Quality**

Although the theoretical material and proof techniques are novel, the paper requires major improvement in writing to be accessible to the RL community. The paper motivates their stochastic approximation scheme by having application in MARL. However, the paper, as it is, is very inaccessible to the RL community.

In many cases, it's better that authors first provide better intuition before jumping into the math and the assumptions.
Many terminologies are assumed to be known by the readers in the RL community, which is not the case. The paper needs to be re-written and be more self-contained so that people don't need to look for the definition of the terminologies in the references.

**Clarity**

The paper needs major improvement in writing.

**Significance**

The paper needs major improvement to be accessible and of interest to the RL community.

**Time Spent Reviewing:**

2

---

> ### Author Response · Authors · 2021-08-10
> **Improving readability and accessibility of the paper**
>
> We thank the reviewer for the comments. To improve readability and accessibility to the RL community, we propose to incorporate the following changes during the review period:
>
> 1. Add additional explanations in Section 4 at the start of each proof.
> 2. Add intuitive explanations for assumptions $A_1 - A_4$ (see also response to Reviewer KYP4, where we have provided an intuitive explanation for $A_2$)
> 3. Unify/simplify terminology across the paper (e.g., Gossip/averaging matrix, “m-fold product”, consensus space, etc.)
> 4. Add some background material to make the text more self-contained (e.g., we will try to explain the ODE approach, martingale difference sequences, globally asymptotically stable equilibrium, Hurwitz-stable matrix, Jordan block/decomposition etc.)
> 5. Add importance of Sec 4.3 explaining its relevance to allow non-square summable step-sizes (see also response to Reviewer KYP4 on this).

---

> > ### Comment · Reviewer_ftij · 2021-08-23
> > **Thanks for the response**
> >
> > I want to thank the authors for providing a plan for improving the readability and accessibility of the paper and also for providing a detailed response to other reviewers. After going through the other reviewers' evaluations and the author's response, I am happy to increase my score to 6.

---

> > > ### Author Response · Authors · 2021-08-24
> > > **Thank you for the positive response**
> > >
> > > Dear Reviewer,
> > >
> > > We thank you for your positive feedback.

---

### Official Review · Reviewer_KYP4 · 2021-07-20

**Rating:** 6
**Confidence:** 3

**Summary:**

This paper studies a family of distributed nonlinear stochastic approximation for MARL, and derives a law of iterated logarithm, which describes the convergence rate on the converging sample path. It also presents an application using TD(0) with linear function approximation, by establishing the convergence rate and showing that it does not depend on the interaction graph.


**Limitations And Societal Impact:**

It would be useful if the authors could provide more intuitive explanation for Assume A2, as well as more concrete application examples.

**Main Review:**

- The problem in consideration is interesting. The paper is also well written and the ideas are carefully presented.
- It would be useful if the authors could provide more intuitive explanation for Assume A2, e.g., what type of scenarios does it imply, or why is it necessary?
- The application example appears to rely on the linear form of h(x). This is consistent with A2. Can this result be generalized to other cases?
- Section 4.3 appears to be technical and not well positioned.
- It would be helpful if the paper could provide an example for the distributed nonlinear SA case? Although Assumption A2 allows nonlinearity (assuming through the two f functions), the application seems to focus on the linear case.


**Time Spent Reviewing:**

3

---

> ### Author Response · Authors · 2021-08-10
> **Intuitive explanation for Assumption A_2 and Examples of distributed nonlinear stochastic approximation**
>
> We sincerely thank the reviewer for the feedback.
>
> $\textbf{Intuitive explanation for Assumption}$ $A_2$ $\textbf{and examples of distributed nonlinear stochastic approximation}$
>
> Eq. (5) in Assumption $A_2$   specifies the structure that $h$ needs to have in a local neighborhood of $x_*$ for Theorem 2.1 to hold. We now provide an intuitive explanation for this decomposition.
>
> First, recall that $\pi$ is the stationary distribution satisfying (3). In particular, it depends on the gossip matrix $W.$ In [26], it was shown that the DSA iterates eventually track the m-fold product of the ODE given in (4). As can be seen, this latter ODE is completely governed by the function $\pi h(x).$ Since $x_*$ is an asymptotically stable equilibrium of this ODE, it readily follows that $\pi h(x_*) = 0.$ However, (5) additionally requires that $\pi h(x)$ be of the form $-\pi (x - x_*)A + \pi f_1(x),$ where $\pi f_1(x)$ represents the higher-order terms in $\pi h(x)$ while $-\pi(x - x_*)A$ represents its linear part with $A$ being positive definite. We emphasize that this is a natural generalization of Assumption (A1) in [33].  The significance of this form of $\pi h(x)$ is that, on the event $\mathcal{E}(x_*),$ the dynamics of the DSA scheme given in (1) will eventually be governed only by this linear term. In fact, one can already see in (15) and (20) that matrix $A$ plays a crucial role in the calculation of our convergence rate.
>
> On the other hand, $(\mathbb{I} - \mathbf{1}'\pi) h(x)$ does not play any role in (4); thus, it has a limited role in the eventual dynamics of the DSA scheme. Consequently, we have minimal assumptions on it. In that, we only require that $(\mathbb{I} - \mathbf{1}'\pi) h(x)$ be of the $(\mathbb{I} - \mathbf{1}' \pi)(B + f_2(x)),$ where $f_2$ is some arbitrary continuous function.
>
> In general, the decomposition given in (5) depends on the distribution $\pi.$ However, there is a family of examples where, surprisingly, this is not true. The simplest member in this family is $h(x) = B − xA,$ where $B$ and $A$ are as described in $A_2.$ Clearly, if $b(i)$ and $x(i)$ are the $i$-th rows of the matrices $B$ and $x,$ respectively, then the $i$-component function here is $h_i(x) = b(i) − x(i)A.$ The fact that the scaling matrix $A$ is the same for each $i$ is crucial for $A_2$ to hold. The other members of the family are obtained by adding various $\pi$-dependent nonlinear perturbations to this simple setup, i.e., by making different choices for $f_1$ and $f_2.$ Footnote 4 on pg. 4 of the supplementary material gives some examples for $f_1$ and $f_2.$ In other words, our results should also be applicable to the distributed TD(0) algorithm when certain nonlinear perturbations get introduced due to interactions between the different agents.
>
> While the scaling matrix $A$ being the same for each $i$ may appear restrictive, we emphasize that this indeed is true in the TD(0) case. In fact, this will be true in the more general cooperative MARL settings as well, as  long as the global state is equally visible to all the agents and only the rewards are local.
>
> $\textbf{Positioning of Section 4.3}$
>
> Our main motivation for adding Section 4.3 was to highlight Lemma 4.8. As emphasized in item 3 of key contributions, Lemma 4.8 is a novel law of iterated logarithm for a sum of scaled martingale differences. Compared to its previous avatar [10, Corollary 6.4.25], this result applies under significantly weaker assumptions. In fact, it is mainly because of this that Lemma 4.4 and, in turn, Theorem 2.1 also hold for non-square summable stepsizes. Not only this, Lemma 4.8 allows even existing results such as the one in [33] to hold for non-square summable stepsizes.
> We will add the above details to better position Section 4.3 and also assist the reader to appreciate the significance of this result in this work.

---

> > ### Comment · Reviewer_KYP4 · 2021-08-24
> > **Thanks for the reply**
> >
> > I thank the authors for the responses. I went through the comments and rebuttals. I would suggest incorporating the more intuition and concrete application examples to the revision.

---

> > > ### Author Response · Authors · 2021-08-24
> > > **Thank you for your feedback**
> > >
> > > Dear Reviewer,
> > >
> > > We sincerely thank you for your constructive feedback. Indeed, we will incorporate your suggestions.

---

### Decision · Program_Chairs · 2021-09-27

**Decision:**

Accept (Poster)

**Comment:**

This paper studies a family of distributed nonlinear stochastic approximation for MARL, and derives a law of iterated logarithm, which describes the convergence rate on the converging sample path. It also presents an application using TD(0) with linear function approximation, by establishing the convergence rate and showing that it does not depend on the interaction graph. All reviewers are convinced on the novelty and contribution of this paper. Reviewers still recommend authors to take their advices in restructuring certain part of the paper to make it more accessible to the general RL community.